# Relationship Between Metabolic Syndrome Indicators Within Reference Ranges and Sarcopenia in Older Women—A 4-Year Longitudinal Study

**DOI:** 10.3390/geriatrics10030076

**Published:** 2025-06-03

**Authors:** Tadayuki Iida, Reina Taguchi, Ruriko Miyashita, Satomi Aoi, Hiromi Ikeda, Nichika Higa, Keiko Kanagawa, Yoko Okuyama, Yasuhiro Ito

**Affiliations:** 1Department of Health and Welfare, Faculty of Health and Welfare, Prefectural University of Hiroshima, Hiroshima 723-0053, Japan; aoi@pu-hiroshima.ac.jp (S.A.); hikeda@pu-hiroshima.ac.jp (H.I.); 2Department Rehabilitation, Hyogo Prefectural Tamba Medical Center, Hyogo 669-3495, Japan; 3Department of Nursing Women’s Health Cares Nursing Master’s Program of Midwifery Course, Kobe City College of Nursing, Kobe 651-2103, Japan; ruriko-miyashita@kobe-ccn.ac.jp; 4Department of Rehabilitation, Yuuaikai Yuuai Medical Center, Social Medical Corporation, Okinawa 901-0224, Japan; 5Graduate Program in Midwifery, Prefectural University of Hiroshima, Hiroshima 723-0053, Japan; k-kanagawa@pu-hiroshima.ac.jp (K.K.); yokuyama@pu-hiroshima.ac.jp (Y.O.); 6Department of Medical Technology, Yokkaichi Nursing and Medical Care University, 1200 Kayocho, Yokkaichi 512-8045, Japan; yasu-ito@y-nm.ac.jp

**Keywords:** sarcopenia, metabolic syndrome indicators, women, reference range

## Abstract

**Background**: Frailty is a state of increased vulnerability to psychosomatic dysfunction associated with aging, with sarcopenia being a major contributing factor. Metabolic-syndrome-related metabolic diseases are recognized as risk factors for sarcopenia. While previous studies have examined the relationship between metabolic disease history or elevated metabolic syndrome indicators and sarcopenia, limited evidence exists regarding the association between metabolic indicators within reference ranges and sarcopenia in the absence of metabolic disease. This study aimed to investigate the relationship between metabolic syndrome indicators within reference ranges and the presence or absence of possible sarcopenia, as well as changes in these indicators over a four-year period, in women aged ≥65 years. **Methods**: A total of 224 community-dwelling women aged ≥65 years from M City and O Town who participated in health check-ups were included (approval no. 20MH017, 1 October 2020). Data were collected on height, body weight, and metabolic indicators (triglycerides, HDL cholesterol, LDL cholesterol, systolic and diastolic blood pressure, and HbA1c) at baseline and after four years. Physical performance was assessed using calf circumference, grip strength, and the five-time sit-to-stand test. Possible sarcopenia was defined according to AWGS2019 criteria. **Results**: Higher baseline HDL cholesterol levels were found to be protective against possible sarcopenia. An increase in triglyceride levels over four years was also associated with a reduced likelihood of possible sarcopenia. **Conclusions**: Maintaining high HDL cholesterol levels and increasing or preserving triglyceride levels may contribute to the prevention of sarcopenia in older women with metabolic indicators within reference ranges.

## 1. Introduction

In 2000, the World Health Organization proposed the idea of healthy life expectancy, and the extension of a healthy lifespan has since been attracting increasing attention. Factors that affect healthy life expectancy include not only diseases and physical health; “frailty” (under the framework of functional health) has also been highlighted. Frailty refers to “a state in which physical dysfunction or health damage may occur as an age-related syndrome” [1]. Frailty is positioned between independence and a condition requiring nursing care, and adequate interventions may lead to recovery from a frail state to independence, preventing a condition requiring nursing care. In Japan, 8.7% of older adults have frailty, while 40.8% have pre-frailty. A previous study reported that the rate of adults with frailty was slightly higher in women, older adults, and those with a poorer socioeconomic or health status [2]. Sarcopenia, characterized by a decrease in skeletal muscle volume, has been identified as an important factor contributing to frailty [3]. According to Amagasa et al., the average intensity of physical activity (METs) was similar between older Japanese men and women. However, women, who generally weigh less than men [4], had lower energy expenditure from physical activity. Since physical activity levels are generally lower in women than in men, causing a decrease in the muscle volume or energy consumption, the control of sarcopenia is an urgent issue in women.

A previous study reported that the concomitant presence of metabolic diseases, such as diabetes mellitus, dyslipidemia, obesity, and osteoporosis, was associated with frailty or the onset of sarcopenia [5]. Caroline et al. cross-sectionally investigated 590 women aged 70 to 79 years and reported a relationship between obesity and physical frailty (sarcopenia) [5]. According to a longitudinal study by Kalyani et al., the incidence of sarcopenia was 3.3-fold higher in a group with HbA1c values ≥8.0% than in a group with HbA1c values <5.5% at baseline, suggesting that a history of diabetes mellitus increases the risk of sarcopenia [6]. Furthermore, after adjustments for sex differences, metabolic syndrome was associated with motor hypofunction, hypo-HDL-cholesterolemia and hyper-triglyceridemia with motor hypofunction, and hypo-HDL-cholesterolemia with reductions in knee extension muscle strength and gait velocity [7,8]. These findings indicate that a relationship exists between metabolic diseases, including obesity-related diabetes mellitus, and sarcopenia. However, these studies investigated the relationship with frailty or sarcopenia in subjects with metabolic diseases at baseline and high-risk subjects based on the reference values of metabolic syndrome indicators. Few studies have examined the risk of frailty or sarcopenia in subjects with metabolic syndrome indicators within their reference ranges. Therefore, it remains unclear how differences in the values of metabolic syndrome indicators within their reference ranges or changes in these values affect frailty or sarcopenia, particularly in individuals with non-metabolic diseases. In the present study, we investigated whether changes in each metabolic syndrome indicator during a four-year period were associated with the presence or absence of the possibility of sarcopenia (possible sarcopenia) in women aged ≥65 years with metabolic syndrome indicators (blood pressure, cholesterol, triglyceride, and HbA1c) within their reference ranges at the start of this survey.

## 2. Materials and Methods

### 2.1. Subjects

The purpose and contents of the present study were explained to participants in special health check-ups in M City and O Town using an explanatory document, and consent regarding participation was obtained from 676 women. In M City, measurements were performed in 2016 (first) and 2020 (four years later). In O town, they were conducted in 2019 (first) and 2023 (four years later). The subjects were 224 women aged ≥65 years with data from two sessions of measurements: at the start of this survey and after four years. From these subjects, we extracted those in whom metabolic syndrome indicators (triglyceride (mg/dL), HDL cholesterol (mg/dL), LDL cholesterol (mg/dL), systolic blood pressure (mmHg), diastolic blood pressure (mmHg), and HbA1c (%)) were within their reference ranges at the start of this survey and performed statistical analyses. A flowchart of the subjects analyzed is shown in Figure 1. The sample size was selected according to diastolic blood pressure and the presence or absence of possible sarcopenia using G-power [9,10]. Based on the findings of the study conducted by Iida T [11] on the relationship between the presence and absence of possible sarcopenia, and in consideration of results showing M1 = 70.0 (mean score of the possible sarcopenia group), SD1 = 13.2, M2 = 77.5 (mean score of the control group), and SD2 = 10.5, as well as one-sided α = 0.05 and power = 80%, the sample size per group was set as 27, for a total sample size of 130 (LDL cholesterol in the smallest number of subjects). Regarding triglyceride (mg/dL), HDL cholesterol (mg/dL), diastolic blood pressure (mmHg), and HbA1c (%), the number of subjects per group was ≥39, meeting the calculated sample size. However, the sample size was not met for LDL cholesterol (mg/dL) or systolic blood pressure (mmHg).

The present study was performed in accordance with the Declaration of Helsinki after its protocol was approved by the Ethics Review Board of the Prefectural University of Hiroshima (approval no. 20MH017, 1 October 2020).

### 2.2. Examination Procedure

Between 2016 and 2023, all subjects were examined twice (at the start of the survey and four years later) for physical characteristics (height, body weight, and BMI) and blood markers of metabolic syndrome. In 2020 and 2023 (four years later), a physical fitness test was performed in addition to these items.

### 2.3. Measurement of Metabolic Syndrome Indicators

Triglyceride (mg/dL), HDL cholesterol (mg/dL), LDL cholesterol (mg/dL), systolic blood pressure (mmHg), diastolic blood pressure (mmHg), and HbA1c (%) were measured as metabolic syndrome indicators. Deteminer C-TG, MetaboLead HDL-C, and MetaboLead LDL-C (Minaris Medical Co., Ltd., Tokyo, Japan) were used as reagents for triglyceride, HDL cholesterol, and LDL cholesterol, respectively. The automatic analyzers Bio Majesty JCA-BM8060 and 9130 (JEOL Ltd., Tokyo, Japan) were used.

Regarding metabolic syndrome indicators, we analyzed subjects in whom the following items were within the reference ranges at the start of this survey: non-fasting triglyceride level ≥ 175 mg/dL, HDL cholesterol level < 40 mg/dL, LDL cholesterol level ≥ 140 mg/dL, systolic blood pressure ≥ 130 mmHg, diastolic blood pressure ≥ 85 mmHg, and HbA1c value of ≥6.1%, which are diagnostic criteria for dyslipidemia, being outside the reference ranges based on the Japan Atherosclerosis Society (JAS) Guidelines for Prevention of Atherosclerotic Cardiovascular Diseases 2022 [12]. Furthermore, in the subjects analyzed, changes during the four-year period were calculated by subtracting the value at the start of this survey from the value for each item after four years.

### 2.4. Criteria for Possible Sarcopenia

A physical fitness test was performed as follows. Calf circumference was measured once on the left side and once on the right side in each subject in a sitting position using a tape measure while exposing the lower legs as extensively as possible. In the grip strength measurement, each subject was instructed to stand with their feet shoulder-width apart and place the upper limbs along the side of the body, with the bilateral elbows extended. In this state, each subject was instructed to grip a dynamometer, and grip strength was measured twice on the left side and twice on the right side. In the five-time sit-to-stand test, each subject was instructed to sit on a 40 cm table and repeat a “standing up and sitting” motion five times in a row while crossing the bilateral upper limbs and holding the shoulders. The time required for five consecutive sessions of motion was measured using a stopwatch. When assessing calf circumference using the AWGS2019 diagnostic criteria for sarcopenia [13], subjects meeting a non-dominant leg calf circumference < 33 cm and (1) a dominant hand grip strength < 18 kg and/or (2) a time ≤ 12 s in the five-time sit-to-stand test were evaluated as having possible sarcopenia.

### 2.5. Statistical Analysis

Age, physical characteristics, and metabolic syndrome factors, including triglycerides, HDL cholesterol, LDL cholesterol, and HbA1c (%), were recorded along with systolic and diastolic pressure. The normality of the initial test values was confirmed using a histogram and the Kolmogorov–Smirnov test (*p* > 0.05). Values from the initial measurements were subtracted from those obtained four years later to assess changes, with normality being tested and confirmed in the same manner (*p* > 0.05). Mean and standard deviation values were calculated for each test item. Furthermore, age, height, weight, BMI, triglycerides, HDL cholesterol, LDL cholesterol, HbA1c, and diastolic and systolic pressure were compared between included and excluded subjects using Student’s *t*-test. Using the results obtained according to criteria for possible sarcopenia as dependent variables, relationships with risk factors were examined using a binomial logistic regression analysis. We assessed the potential for multicollinearity between age, BMI, and the various metabolic syndrome indicators (triglyceride, HDL cholesterol, LDL cholesterol, systolic blood pressure, diastolic blood pressure, and HbA1c) using variance inflation factors (VIFs). A VIF value of less than 10 was considered indicative of no significant multicollinearity. All VIFs in our models were found to be less than 10, suggesting that the estimated regression coefficients were not unduly inflated by collinearity. The explanatory variables were triglyceride (mg/dL), HDL cholesterol (mg/dL), LDL cholesterol (mg/dL), systolic blood pressure (mmHg), diastolic blood pressure (mmHg), and HbA1c (%) values at the start of this survey. Age and BMI at the start of this survey were adjusted. In addition, changes in the above values were used. Multicollinearity was assessed for all independent variables, including baseline age, baseline BMI, baseline metabolic syndrome indicators (triglyceride, HDL cholesterol, LDL cholesterol, systolic blood pressure, diastolic blood pressure, and HbA1c), and their respective four-year change values. VIFs were computed, and all values were found to be less than 10. This confirms that these variables did not exhibit problematic collinearity, allowing for the robust estimation of their associations with outcomes over time. Age, BMI, and the value for each item at the start of this survey were adjusted. In all tests, a *p*-value ≤ 0.05 was regarded as significant. EZR (version 1.68) was used for statistical analyses.

## 3. Results

### 3.1. Comparisons of Physical Characteristics and Metabolic Syndrome Indicators

Comparisons of physical characteristics and metabolic syndrome indicators (triglyceride, HDL cholesterol, LDL cholesterol, HbA1c, maximum blood pressure, and minimum blood pressure) at the start of this survey and after four years in the subjects analyzed are shown in Table 1. HDL cholesterol, maximum blood pressure, minimum blood pressure, and HbA1c values were significantly higher after four years than at the start of this survey, while height was significantly lower.

### 3.2. Relationship Between Baseline Metabolic Syndrome Indicators and the Presence or Absence of Possible Sarcopenia

The results of the logistic regression analysis using the presence or absence of possible sarcopenia as a response variable are shown in Table 2. Metabolic syndrome indicators at the start of this survey were associated with the presence or absence of possible sarcopenia after four years. The HDL cholesterol level at the start of this survey was correlated with the presence or absence of possible sarcopenia (odds ratio = 0.966; *p*-value = 0.022). Blood pressure, LDL cholesterol level, triglyceride level, and HbA1c value at the start of this survey did not correlate with the presence or absence of possible sarcopenia.

### 3.3. Relationship Between Changes in Metabolic Syndrome Indicators over Four Years and the Presence or Absence of Possible Sarcopenia

Table 3 shows the impact of changes in metabolic syndrome indicators during the four-year period on the presence or absence of possible sarcopenia. A change in the triglyceride level between measurements at the start of this survey and after four years correlated with the presence or absence of possible sarcopenia (odds ratio = 0.987; *p*-value = 0.039). However, changes in blood pressure, HDL cholesterol, LDL cholesterol, and HbA1c did not correlate with the presence or absence of possible sarcopenia.

## 4. Discussion

These results show that a high HDL cholesterol level at the start of the survey was a protective factor for the possibility of sarcopenia. According to the JAS Guidelines for Prevention of Atherosclerotic Cardiovascular Diseases 2022, a diagnosis of dyslipidemia is reached based on the following criteria: triglyceride, ≥150 mg/dL after fasting or ≥175 mg/dL under a non-fasted condition; HDL cholesterol, <40 mg/dL; LDL cholesterol, ≥140 mg/dL; and non-HDL cholesterol, 170 mg/dL [12]. Briefly, in the presence of a decrease in HDL cholesterol level, a diagnosis of dyslipidemia is made. Bi et al. reported that sarcopenia was associated with the presence or absence of dyslipidemia [14]. This finding may be attributed to age-related fat inflammation inducing the redistribution of adipose tissue into the abdominal cavity, resulting in skeletal muscle lipid infiltration and a decline in physical activities [15]. Lipids and their derivatives may have accumulated in and around myocytes, inducing mitochondrial dysfunction, hindering the β-oxidation of fatty acids, enhancing the production of active oxygen, causing insulin resistance and lipotoxicity, and promoting the secretion of some proinflammatory cytokines [16]. In addition, these cytokines may have induced inflammation and inhibited a decrease in adipose tissue, establishing a vicious circle consisting of systemic local hyperlipidemia, inflammation, and insulin resistance, thereby inducing sarcopenia [17]. Collectively, these findings suggest the importance of preventing a decrease in the HDL cholesterol level, as a diagnostic criterion for dyslipidemia, and maintaining it within its reference range for the prevention of sarcopenia, because dyslipidemia leads to a decrease in skeletal muscle volume.

Furthermore, previous studies have suggested that hypo-HDL-cholesterolemia, in which the HDL cholesterol level decreases due to a lack of exercise, obesity, and smoking, is associated with motor hypofunction [7,18]. Park et al. reported that the HDL cholesterol level was high when there was an increase in physical activity, which decreased the prevalence of sarcopenia [19]. Therefore, sufficient exercise is necessary to achieve an increase in HDL cholesterol level. However, Grimmer et al. showed that physical activity levels were significantly lower in women than in men [4]. Furthermore, Amagasa et al. demonstrated that the average intensity of physical activity (METs) was similar between older Japanese men and women. However, women, who generally weigh less than men [20], had lower energy expenditure from physical activity. These findings suggest that activity levels and energy consumption are generally lower in women than in men, and also that a decrease in activity levels often leads to a reduction in the HDL cholesterol level; therefore, increases in physical activities may be necessary in women, more so than in men. According to Xue et al., a reduction in activity levels decreased energy consumption, leading to weight loss through anorexia/reduced dietary intake and resulting in sarcopenia [21]. In addition, sarcopenia reduces vitality, muscle strength, and physical functions, contributing to a cycle that promotes a reduction in activity levels [21]. Therefore, a low HDL cholesterol level may be avoided by preventing motor hypofunction or a reduction in activity levels, thereby preventing sarcopenia.

In this study, HDL cholesterol levels were associated with the onset of sarcopenia, which may also be influenced by hormonal factors. The rapid decline in estrogen levels during menopause, along with their persistently lower levels compared to men, has been suggested to contribute to decreased bone mineral density due to increased bone resorption, lower HDL cholesterol levels [22], and vascular endothelial dysfunction [23]. In addition, Chamberlain reported that a decrease in estrogen level affected multichain lipase activity, decreasing the synthesis of HDL cholesterol [24]. According to Huang et al., a decrease in estrogen level may be associated with increases in the levels of THF-α, IL-6, and other inflammatory factors, which may subsequently decrease muscle mass [25]. Therefore, a postmenopausal decrease in estrogen level may lead to a reduced HDL cholesterol level, increasing the risk of sarcopenia through a decrease in bone mineral density or muscle volume. HDL cholesterol has been shown to be positively correlated with skeletal muscle mass, and individuals with greater muscle mass tend to exhibit higher levels of HDL cholesterol [26]. Elevated HDL cholesterol levels have also been associated with shorter recovery times of skeletal muscle enzymes, suggesting improved mitochondrial function and metabolic efficiency in skeletal muscle [27]. Therefore, higher HDL cholesterol levels may reflect enhanced physical performance and functional capacity [27]. Conversely, extremely high concentrations of HDL cholesterol (≥70 mg/dL) have been reported to be potentially associated with an increased risk of muscle strength decline and the development of sarcopenia [28]. In that study, adjustments for body mass index (BMI), a known factor influencing both sarcopenia and muscle strength [29], were not performed. The present study differs in that BMI was accounted for in the analysis in addition to age. Notably, our findings provide novel evidence that elevated HDL cholesterol levels within the normal range (≥40 mg/dL) in older women may be protective against the development of sarcopenia. Based on the results of this study, it is important to maintain the HDL cholesterol level in postmenopausal adult women, which prevents the transition to sarcopenia.

The changes observed in the triglyceride level during the four-year period showed that it functioned as a protective factor for sarcopenia when it was higher after four years. Jiang et al. suggested that increases in lipid-metabolism-associated parameters (BMI, triglyceride, total cholesterol, and LDL cholesterol) within their reference ranges prevent sarcopenia [30]. In addition, Perna et al. conducted a study on older adults and indicated that increases in obesity and metabolic parameters within their reference ranges may help prevent muscle loss [31]. Jiang et al. also demonstrated that the prevalence of sarcopenia decreased with increases in body fat percentage [30]. Taken together, these findings suggest that maintaining or increasing triglyceride levels within their reference ranges, without reducing percent body fat, may help prevent sarcopenia and muscle loss in older adults. Furthermore, Norman found that age-related anorexia may lead to eating disorders or reduced dietary intake, resulting in malnutrition [32]. Rondanelli and colleagues demonstrated that the consumption of Cynara scolymus extract in overweight individuals diagnosed with impaired fasting glucose (IFG) led to improvements in both glucose metabolism and various parameters of the lipid profile (specifically total cholesterol, LDL-C, HDL-C, triglycerides, and apolipoprotein B) [33]. Similarly, Ezaki et al. investigated the impact of medium-chain triglyceride (MCT) consumption on muscle mass and function in frail older adults, reporting significant improvements in muscle mass, strength, and walking speed [34]. While these findings demonstrate improvements through interventional studies, they also imply that nutritional deficiencies could potentially lead to abnormal glucose and lipid metabolism in older adults. Since excessive malnutrition reduces the triglyceride level, nutritional management may be important to maintain the triglyceride level within its reference range. Therefore, the sufficient consumption of meals and the prevention of an excessive decrease in triglyceride level may lead to the prevention of sarcopenia.

Pyka et al. make important recommendations regarding obesity and weight loss in the elderly [35]. They state that weight loss should be recommended for obese older adults with evidence of functional or metabolic impairment, suggesting that weight loss may improve such impairment. At the same time, they caution that unintentional weight loss in the elderly poses significant health risks and that weight loss should be accompanied by the careful monitoring of muscle mass and bone density. In the present study, participants exhibited an average weight loss of 0.5 kg over a four-year period. Considering that the average body weight of Japanese individuals is substantially lower than that of older adults in the United States (e.g., 78.2 kg among women aged 65 years and older), it is likely that few participants met the criteria for obesity with functional impairment as described by Pyka et al. Therefore, it is plausible that some participants experienced unintentional weight loss. Therefore, it cannot be ruled out that unintended weight loss may have reduced muscle mass and bone density and included subjects with suspected sarcopenia.

There were several limitations in the present study. First, our sample was limited to women of a geographically localized area (M City and O Town), and thus it would be inappropriate to generalize the findings, although the subjects’ heights and weights were similar to those in a Japanese national survey [36]. In addition, the women’s lifestyles, being geographically localized (M City and O Town), were likely to be uniform, unlike lifestyles in the general population, and while this homogeneity was likely to have contributed to the internal validity of the results, it also make our findings less generalizable. The subjects were not patients at medical institutions or facility residents, but free-will-based participants in the health survey; therefore, they may have been biased to good health or very health-conscious individuals. Furthermore, dietary assessments, physical activity evaluations, and measurements of muscle mass and strength—which could potentially influence sarcopenia—were not conducted in this study. Therefore, the heterogeneity in lifestyle factors cannot be ruled out. In addition, hormonal therapy was not investigated in this study. Salpeter et al. reported the risk of coronary heart disease (CHD) in women aged 60 years or older or more than 10 years postmenopausal, with and without hormone replacement therapy (HRT) [37]. Their findings demonstrated that the relative risk of CHD in women receiving HRT compared to those not receiving it was 1.03, indicating that HRT had no significant impact on CHD incidence in this population. Additionally, in Japan, the prevalence of HRT use among women aged 45–64 years is reported to be 2.5% [38]. In our study, the mean age of participants at baseline was 70.2 years, and applying the same 2.5% prevalence, this would correspond to only six individuals in the analysis cohort. Therefore, the influence of HRT or related hormonal treatments is considered to be minimal. Second, possible sarcopenia was determined based solely on the AWGS2019 sarcopenia diagnostic criteria. As there were no confirmatory diagnoses from specialist physicians diagnosing older people, false-positive or -negative sarcopenia might have been present, which could have distorted the association. In addition, the possibility of sarcopenia was not investigated at the start of this survey, so we cannot rule out possible sarcopenia already being present at the start of this survey. Third, in the examinations of LDL cholesterol (mg/dL) and systolic blood pressure (mmHg), the sample size was small and so there may be a deviation in the results obtained. Moreover, metabolic syndrome indicators were measured in a non-fasting state. Stricter cutoff values were applied at baseline compared to those used under fasting conditions, and thus the classification of participants at baseline is considered to have been appropriately conducted. However, in terms of changes over time, these indicators may be influenced not only by dietary intake but also by physical activity and alcohol consumption, which could have introduced discrepancies in the observed results. These issues warrant further study. Dzięgielewska-Gęsiak et al. divided their study into high- and low-HDL-C groups and found that the high-HDL-C group had a smaller waist circumference and lower triglyceride concentrations [39]. Although that was a cross-sectional study, the results are considered to support the findings of the present study. Therefore, the results of this four-year longitudinal survey showed for the first time that HDL cholesterol levels were associated with possible sarcopenia.

## 5. Conclusions

The present study demonstrated that a high HDL cholesterol level at the start of this survey prevented sarcopenia in healthy women aged ≥65 years. In addition, changes in triglyceride level over the four-year study period revealed that a higher value after four years contributed to the prevention of sarcopenia. This result newly suggests that a high HDL cholesterol level and changes in triglyceride level are parameters for the prevention of sarcopenia in individuals with metabolic syndrome indicators within their reference ranges.

## Figures and Tables

**Figure 1 geriatrics-10-00076-f001:**
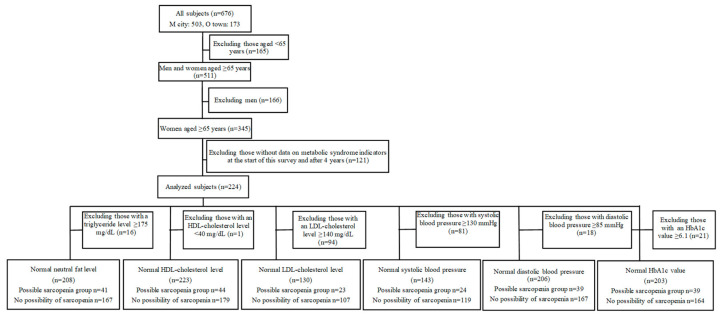
Flowchart of the analyzed subjects.

**Table 1 geriatrics-10-00076-t001:** Physical characteristics and metabolic syndrome indicators of study participants.

	Baseline		4 Years Later		
	Mean	(SD)	Mean	(SD)	*p*-Value
Age (years)	70.2	(5.9)	74.2	(5.9)	<0.001
Height (cm)	151.5	(5.8)	150.6	(6.1)	<0.001
Weight (kg)	51.3	(7.4)	50.8	(7.4)	0.004
BMI (kg/m^2^)	22.4	(3.1)	22.4	(3.2)	0.556
Triglycerides (mg/dL)	101.8	(47.4)	102.4	(51.5)	0.832
HDL cholesterol (mg/dL)	67.1	(14.9)	72.0	(16.6)	<0.001
LDL cholesterol (mg/dL)	132.3	(29.8)	127.7	(32.8)	0.026
Systolic blood pressure (mmHg)	124.4	(14.4)	134.3	(18.2)	<0.001
Diastolic blood pressure (mmHg)	71.8	(8.9)	74.6	(9.6)	<0.001
HbA1c (%)	5.5	(0.4)	5.7	(0.5)	<0.001

**Table 2 geriatrics-10-00076-t002:** Relationship between baseline metabolic syndrome indicators and the presence or absence of possible sarcopenia at baseline.

	Odds Ratio	95%Cl			*p*-Value
Triglycerides (mg/dL) *^1^	1.010	0.999	-	1.030	0.070
HDL cholesterol (mg/dL) *^1^	0.966	0.938	-	0.995	0.022
LDL cholesterol (mg/dL) *^1^	1.010	0.977	-	1.040	0.582
Systolic blood pressure (mmHg) *^1^	0.968	0.920	-	1.020	0.198
Diastolic blood pressure (mmHg) *^1^	0.993	0.947	-	1.040	0.783
HbA1c (%) *^1^	3.230	0.686	-	15.200	0.138

*^1^ A logistic regression analysis of each metabolic syndrome indicator adjusted for baseline age and BMI.

**Table 3 geriatrics-10-00076-t003:** Relationship between changes in metabolic syndrome indicators over four years and the presence or absence of sarcopenia.

	Odds Ratio	95%Cl			*p*-Value
Change in triglycerides (mg/dL) *^2^	0.987	0.975	-	0.999	0.039
Change in HDL cholesterol (mg/dL) *^3^	1.010	0.974	-	1.060	0.481
Change in LDL cholesterol (mg/dL) *^4^	1.000	0.984	-	1.030	0.661
Change in systolic blood pressure (mmHg) *^5^	0.960	0.915	-	1.010	0.093
Change in diastolic blood pressure (mmHg) *^6^	0.981	0.951	-	1.010	0.238
Change in HbA1c (%) *^7^	0.232	0.035	-	1.550	0.132

*^2^ A logistic regression analysis of changes in each metabolic syndrome indicator, adjusted for age, BMI, and the baseline triglycerides. *^3^ A logistic regression analysis of changes in each metabolic syndrome indicator, adjusted for age, BMI, and the baseline HDL cholesterol. *^4^ A logistic regression analysis of changes in each metabolic syndrome indicator, adjusted for age, BMI, and the baseline LDL cholesterol. *^5^ A logistic regression analysis of changes in each metabolic syndrome indicator, adjusted for age, BMI, and the baseline systolic blood pressure. *^6^ A logistic regression analysis of changes in each metabolic syndrome indicator, adjusted for age, BMI, and the baseline diastolic blood pressure. *^7^ A logistic regression analysis of changes in each metabolic syndrome indicator, adjusted for age, BMI, and the baseline HbA1c.

## Data Availability

Obtained data cannot be shared publicly because the datasets have ethical or legal restrictions for public deposition owing to inclusion of sensitive information from human participants. Based on regulations regarding ethical guidelines in Japan, the ethical review board of the Faculty of Health and Welfare, Prefectural University of Hiroshima, imposed restrictions on the data collected in this study. The data that support the findings of this study are available from the corresponding author upon reasonable request.

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
