# Peer review of "Relationship Between Metabolic Syndrome Indicators Within Reference Ranges and Sarcopenia in Older Women—A 4-Year Longitudinal Study"

_geriatrics, 2025, doi:10.3390/geriatrics10030076_

Round 1
Reviewer 1 Report
Comments and Suggestions for Authors
The study addresses an important gap by examining the relationship between metabolic indicators within reference ranges and sarcopenia risk in older women. However I have some suggestions:
- Participants were volunteers attending health check-ups, possibly representing a healthier subset of the population, which may limit generalizability.
- The study population is geographically localized (M City and O Town), which may limit applicability to broader or more diverse populations.
- The study did not assess sarcopenia at baseline, making it difficult to determine incident cases or establish temporal causality.
- Lack of detailed data on diet, physical activity, smoking, and other lifestyle factors that influence metabolic health and muscle mass - see: DOI: https://doi.org/10.18794/aams/109954 and if applicable add into introduction or discussion.
- Although calculations were performed, some analyses (e.g., LDL cholesterol, systolic blood pressure) did not meet the desired sample size, potentially limiting statistical power and robustness. See also: doi:10.20 452/pamw.3559
- Some metabolic indicators (e.g., triglycerides, glucose) were measured in non-fasted states, which can introduce variability.
- Hormonal status (e.g., menopause, hormone therapy), medication use, and other health conditions were not controlled for, which could influence results.
- Adjustments were made for age and BMI, but other factors influencing sarcopenia risk may not have been fully accounted for - please add and discuss
Author Response
Dear Reviewer 1
We are grateful to reviewer 1 for the critical comments and useful suggestions that have helped us to improve our manuscript. As indicated in the responses that follow, we have taken all these comments and suggestions into account in the revised version of our manuscript. Slight improvements in English and/or figures to clearly communicate my research, as pointed out by the reviewers, will be addressed by my request to MDPI's Author Services after the reviewers have seen this revised manuscript.
Comments 1: Participants were volunteers attending health check-ups, possibly representing a healthier subset of the population, which may limit generalizability.
Comments 2: The study population is geographically localized (M City and O Town), which may limit applicability to broader or more diverse populations.
Reply 1 and 2
P9 L330 - 342
Thank you for pointing out this important point. I added some thoughts to the paragraph on limitations in discussion.
“There were several limitations in the present study. First, our sample was limited to women of geographically localized (M City and O Town), and thus it would be inappropriate to generalize the findings, although the subjectʼ height and weight were similar to those of a Japanese national survey [30]. In addition, the women’s lifestyles of geographically localized (M City and O Town) were likely to be uniform, unlike lifestyles in the general population, and while this homogeneity was likely to have contributed to the internal validity of the results, it also make our findings less generalizable. Subjects were not patients at medical institutions or facility residents, but free will-based participants in the health survey; therefore, they may have been biased to good health or very health-conscious individuals. Furthermore, dietary assessments, physical activity evaluations, and measurements of muscle mass and strength—which could potentially influence sarcopenia—were not conducted in this study. Therefore, the heterogeneity in lifestyle factors cannot be ruled out.”
Comments 3: The study did not assess sarcopenia at baseline, making it difficult to determine incident cases or establish temporal causality.
Reply 3
P9 L351-355
Thank you for pointing out this important point. I added some thoughts to the paragraph on limitations in discussion.
“Second, the possible sarcopenia was determined based solely on the AWGS2019 sarcopenia diagnostic criteria. As there were no confirmatory diagnoses from specialist physicians diagnosing older people, false-positive or -negative sarcopenia might have been present, which could have distorted the association. And the possibility of sarcopenia was not investigated at the start of this survey, we cannot rule out possible sarcopenia already being present at the start of this survey. “
Comments 4: Lack of detailed data on diet, physical activity, smoking, and other lifestyle factors that influence metabolic health and muscle mass - see: DOI: https://doi.org/10.18794/aams/109954 and if applicable add into introduction or discussion.
Reply 4
P8 L316-329
A new paragraph has been added to the discussion.
“Pyka et al. make important recommendations regarding obesity and weight loss in the elderly. They state that weight loss should be recommended for obese older people with evidence of functional or metabolic impairment, suggesting that weight loss may improve such impairment. At the same time, they caution that unintentional weight loss in the elderly poses significant health risks and that weight loss should be accompanied by careful monitoring of muscle mass and bone density. In the present study, participants exhibited an average weight loss of 0.5 kg over a four-year period. Considering that the average body weight of Japanese individuals is substantially lower than that of older adults in the United States (e.g., 78.2 kg among women aged 65 years and older), it is likely that few participants met the criteria for obesity with functional impairment as described by Pyka et al. Therefore, it is plausible that some participants experienced unintentional weight loss. Therefore, it cannot be ruled out that unintended weight loss may have reduced muscle mass and bone density and included subjects with suspected sarcopenia caused.”
Comments 5: Although calculations were performed, some analyses (e.g., LDL cholesterol, systolic blood pressure) did not meet the desired sample size, potentially limiting statistical power and robustness. See also: doi:10.20 452/pamw.3559
Reply 5
P9 L365-367
Thank you for providing the references. Although our sample size was small and the results may be uncertain, the presented papers support the findings of the study. Therefore, we have added the following:
“DziÄ™gielewska-GÄ™siak et al. divided the study into high and low HDL-C groups and found that the high HDL-C group had a smaller waist circumference and lower triglyceride concentrations. Although this is a cross-sectional study, the results are considered to support the findings of the present study.”
Comments 6: Some metabolic indicators (e.g., triglycerides, glucose) were measured in non-fasted states, which can introduce variability.
Reply 6
P9 L359-365
In the present study, we utilized a non-fasting cutoff value of 175 mg/dL. Considering that the standard cutoff under fasting conditions is 150 mg/dL, we believe that the classification at baseline was conducted appropriately. However, as you have thoughtfully pointed out, the four-year changes in triglyceride levels may have been influenced by various factors, including physical activity, dietary intake, and alcohol consumption. In response to your valuable comment, we have added this point to the Limitation section of the manuscript.
“Also, metabolic syndrome indicators were measured in a non-fasting state. Stricter cutoff values were applied at baseline compared to those used under fasting conditions, and thus the classification of participants at baseline is considered to have been appropriately conducted. However, in terms of changes over time, these indicators may be influenced not only by dietary intake but also by physical activity and alcohol consumption, which could have introduced discrepancies in the observed results.”
Comments 7: Hormonal status (e.g., menopause, hormone therapy), medication use, and other health conditions were not controlled for, which could influence results.
Reply 7
P9 L342-351
Hormonal therapy was not investigated in this study. The following was added. However, we acknowledge that potential confounding factors such as physician diagnoses (section 3) and changes in metabolic syndrome indicators due to physical activity or dietary habits (section 6) cannot be entirely ruled out. As such, these points have been added to the Limitation section of the manuscript.
“Hormonal therapy was not investigated in this study. Salpeter et al. [1] reported the risk of coronary heart disease (CHD) in women aged 60 years or older or more than 10 years postmenopausal, with and without hormone replacement therapy (HRT). Their findings demonstrated that the relative risk of CHD in women receiving HRT compared to those not receiving it was 1.03, indicating that HRT had no significant impact on CHD incidence in this population. Additionally, in Japan, the prevalence of HRT use among women aged 45–64 years is reported to be 2.5% [2]. In our study, the mean age of participants at baseline was 70.2 years, and applying the same 2.5% prevalence, this would correspond to only six individuals in the analysis cohort. Therefore, the influence of HRT or related hormonal treatments is considered to be minimal.”
[1] Salpeter SR, Walsh JM, Greyber E, Salpeter EE. Brief report: Coronary heart disease events associated with hormone therapy in younger and older women. A meta-analysis. J Gen Intern Med. 2006 Apr;21(4):363-6. doi: 10.1111/j.1525-1497.2006.00389.x.
[2] Nagata C, Matsushita Y, Shimizu H. Prevalence of hormone replacement therapy and user's characteristics: a community survey in Japan. Maturitas. 1996 Nov;25(3):201-7. doi: 10.1016/s0378-5122(96)01067-5.
Comments 8: Adjustments were made for age and BMI, but other factors influencing sarcopenia risk may not have been fully accounted for - please add and discuss
Reply 8
As you point out, and sarcopenia includes diet, exercise, muscle mass and strength. These were not investigated in this study and lifestyle heterogeneity cannot be ruled out. This research limitation is included in section 1.
Reviewer 2 Report
Comments and Suggestions for Authors
This is an interesting paper. I suggest few modifications:
Add a brief discussion section that explores plausible physiological mechanisms linking HDL cholesterol and triglycerides to muscle health. This strengthens the scientific context and provides a rationale for future research.
In the Methods section, clearly specify which variables were included as covariates in the analysis (e.g., age, BMI, physical activity level, etc.). Also, clarify whether multiple comparisons were addressed, especially given the number of metabolic indicators tested.
Cite the paper by Rondanelli et al. (2013) titled: "Metabolic management in overweight subjects with naive impaired fasting glycaemia by means of a highly standardized extract from cynara scolymus: A double-blind, placebo-controlled, randomized clinical trial"
You can cite as follow
you can include it in your paper where you discuss metabolic regulation , impaired fasting glucose , or nutritional interventions aimed at improving metabolic parameters—especially if you're discussing how metabolic health impacts aging or sarcopenia.
Author Response
Dear Reviewer 2
We are grateful to reviewer 2 for the critical comments and useful suggestions that have helped us to improve our manuscript. As indicated in the responses that follow, we have taken all these comments and suggestion into account in the revised version of our manuscript. Slight improvements in English and/or figures to clearly communicate my research, as pointed out by the reviewers, will be addressed by my request to MDPI's Author Services after the reviewers have seen this revised manuscript.
Comments 1: Add a brief discussion section that explores plausible physiological mechanisms linking HDL cholesterol and triglycerides to muscle health. This strengthens the scientific context and provides a rationale for future research.
Reply 1
P7 L275-P8 287
Thank you for pointing out this important point. I added some thoughts to the paragraph on HDL.
“HDL cholesterol has been shown to be positively correlated with skeletal muscle mass, and individuals with greater muscle mass tend to exhibit higher levels of HDL cholesterol [1]. Elevated HDL cholesterol levels have also been associated with shorter recovery times of skeletal muscle enzymes, suggesting improved mitochondrial function and metabolic efficiency in skeletal muscle [2]. Therefore, higher HDL cholesterol levels may reflect enhanced physical performance and functional capacity [2]. Conversely, extremely high concentrations of HDL cholesterol (≥70 mg/dL) have been reported to be potentially associated with an increased risk of muscle strength decline and the development of sarcopenia [3]. In the study, adjustments for body mass index (BMI), a known factor influencing both sarcopenia and muscle strength [4], were not performed. In contrast, the present study differs in that BMI, in addition to age, was accounted for in the analysis. Notably, our findings provide novel evidence that elevated HDL cholesterol levels within the normal range (≥40 mg/dL) in older women may be protective against the development of sarcopenia."
- Skeletal Muscle Mass is Associated with HDL Cholesterol Levels and the Ratio of LDL to HDL Cholesterol in Young Men: A Pilot Study
- HDL-C and apolipoprotein A-I are independently associated with skeletal muscle mitochondrial function in healthy humans
- High-density lipoprotein cholesterol level and risk of muscle strength decline: A longitudinal study
- Relationship of Fat Mass Index and Fat Free Mass Index With Body Mass Index and Association With Function, Cognition and Sarcopenia in Pre-Frail Older Adults
Comments 2: In the Methods section, clearly specify which variables were included as covariates in the analysis (e.g., age, BMI, physical activity level, etc.). Also, clarify whether multiple comparisons were addressed, especially given the number of metabolic indicators tested.
Reply 2
Each explanatory variable is clearly stated and a note is added on tests of collinearity. We have also added details to Table 3.
P4 L171- L176
We assessed the potential for multicollinearity between age, BMI, and the various metabolic syndrome indicators (triglyceride, HDL cholesterol, LDL cholesterol, systolic blood pressure, diastolic blood pressure, HbA1c) using Variance Inflation Factors (VIFs). A VIF value less than 10 was considered indicative of no significant multicollinearity. All VIFs in our models were found to be less than 10, suggesting that the estimated regression coefficients were not unduly inflated by collinearity.
P4 L180-P5 186
Multicollinearity was assessed for all independent variables, including baseline age, baseline BMI, baseline metabolic syndrome indicators (triglyceride, HDL cholesterol, LDL cholesterol, systolic blood pressure, diastolic blood pressure, HbA1c), and their respective 4-year change values. VIFs were computed, and all values were found to be less than 10. This confirms that these variables did not exhibit problematic collinearity, allowing for robust estimation of their associations with outcomes over time. 
Comments 3: Cite the paper by Rondanelli et al. (2013) titled: "Metabolic management in overweight subjects with naive impaired fasting glycaemia by means of a highly standardized extract from cynara scolymus: A double-blind, placebo-controlled, randomized clinical trial"
You can cite as follow
you can include it in your paper where you discuss metabolic regulation , impaired fasting glucose , or nutritional interventions aimed at improving metabolic parameters—especially if you're discussing how metabolic health impacts aging or sarcopenia.
Reply 3
Thank you for sharing this important literature with us. Thank you for pointing out this important point. I added some thoughts to the paragraph on triglyceride.
P8 L302- L311
“Rondanelli and colleagues [1] demonstrated that the consumption of Cynara scolymus extract in overweight individuals diagnosed with impaired fasting glucose (IFG) led to improvements in both glucose metabolism and various parameters of the lipid profile (specifically total cholesterol, LDL-C, HDL-C, triglycerides, and apolipoprotein B). Similarly, Ezaki et al. [2] investigated the impact of medium-chain triglyceride (MCT) consumption on muscle mass and function in frail older adults, reporting significant improvements in muscle mass, strength, and walking speed. While these findings demonstrate improvements through interventional studies, they also infer that nutritional deficiencies could potentially lead to abnormal glucose and lipid metabolism in older adults.”
- Metabolic management in overweight subjects with naive impaired fasting glycaemia by means of a highly standardized extract from cynara scolymus: A double-blind, placebo-controlled, randomized clinical trial
- Medium-chain triglycerides (8:0 and 10:0) increase muscle mass and function in frail older adults: a combined data analysis of clinical trials
Round 2
Reviewer 1 Report
Comments and Suggestions for Authors
Table 2 title should be corrected into
Table 2 Relationship between baseline metabolic syndrome indicators and the presence or absence of sarcopenia at baseline
Author Response
Dear Reviewer 1
We received useful suggestions from reviewer 1, which we used to improve the manuscript. As indicated in the response below, we have taken your comments into account and prepared a revised manuscript. The revised sections are indicated by yellow markers.
Comments 1: Table 2 title should be corrected into
Reply 1:
P5 L213
In accordance with your suggestion, the title of Table 2 has been amended as follows.
“Table 2 Relationship between baseline metabolic syndrome indicators and the presence or absence of possible sarcopenia at baseline”